# Anti-Diabetic Therapy, Heart Failure and Oxidative Stress: An Update

**DOI:** 10.3390/jcm11164660

**Published:** 2022-08-09

**Authors:** Ioanna Koniari, Dimitrios Velissaris, Nicholas G. Kounis, Eleni Koufou, Eleni Artopoulou, Cesare de Gregorio, Virginia Mplani, Themistoklis Paraskevas, Grigorios Tsigkas, Ming-Yow Hung, Panagiotis Plotas, Vaia Lambadiari, Ignatios Ikonomidis

**Affiliations:** 1Department of Cardiology, University Hospital of South Manchester NHS Foundation Trust, Manchester M23 9LT, UK; 2Department of Internal Medicine, University Hospital of Patras, 26500 Patras, Greece; 3Department of Cardiology, University Hospital of Patras, 26500 Patras, Greece; 4Department of Clinical and Experimental Medicine, University of Messina Medical School, 98122 Messina, Italy; 5Intensive Care Unit, Patras University Hospital, 26500 Patras, Greece; 6Division of Cardiology, Department of Internal Medicine, Shuang Ho Hospital, Taipei Medical University, New Taipei City 23561, Taiwan; 7Division of Cardiology, Department of Internal Medicine, School of Medicine, College of Medicine, Taipei Medical University, Taipei 11031, Taiwan; 8Taipei Heart Institute, Taipei Medical University, Taipei 11031, Taiwan; 9Laboratory Primary Health Care, School of Health Rehabilitation Sciences, University of Patras, 26504 Patras, Greece; 10Second Department of Internal Medicine, Attikon University Hospital, Medical School, National and Kapodistrian University of Athens, 12462 Athens, Greece; 11Second Cardiology Department, Attikon University Hospital, Medical School, National and Kapodistrian University of Athens, 12462 Athens, Greece

**Keywords:** diabetes, heart failure with preserved ejection fraction (HFpEF), heart failure with reduced ejection fraction (HFrEF), oxidative stress, metformin, SGLT2 inhibitors, thiazolidinediones, sulfonylureas, dipeptidyl peptidase 4 inhibitors, glucagon-likepeptide-1 receptor agonists

## Abstract

Diabetes mellitus (DM) and heart failure (HF) are two chronic disorders that affect millions worldwide. Hyperglycemia can induce excessive generation of highly reactive free radicals that promote oxidative stress and further exacerbate diabetes progression and its complications. Vascular dysfunction and damage to cellular proteins, membrane lipids and nucleic acids can stem from overproduction and/or insufficient removal of free radicals. The aim of this article is to review the literature regarding the use of antidiabetic drugs and their role in glycemic control in patients with heart failure and oxidative stress. Metformin exerts a minor benefit to these patients. Thiazolidinediones are not recommended in diabetic patients, as they increase the risk of HF. There is a lack of robust evidence on the use of meglinitides and acarbose. Insulin and dipeptidyl peptidase-4 (DPP-4) inhibitors may have a neutral cardiovascular effect on diabetic patients. The majority of current research focuses on sodium glucose cotransporter 2 (SGLT2) inhibitors and glucagon-like peptide 1 (GLP-1) receptor agonists. SGLT2 inhibitors induce positive cardiovascular effects in diabetic patients, leading to a reduction in cardiovascular mortality and HF hospitalization. GLP-1 receptor agonists may also be used in HF patients, but in the case of chronic kidney disease, SLGT2 inhibitors should be preferred.

## 1. Introduction

Diabetes mellitus (DM) and heart failure (HF) are interconnected diseases [1]. It is widely known that the prevalence of HF is higher in patients with DM, while the prognosis of these patients is worse compared to HF patients without diabetes [1]. The fact that HF is present in patients with diabetes even in the absence of coronary artery disease (CAD), valvular disease and any other cardiovascular risk factors has led to the introduction of the term “diabetic cardiomyopathy” to address the need for better glycemic control in these patients [1,2]. Endothelial dysfunction and accelerated atherosclerosis constitute leading causes of morbidity and mortality associated with DM. The excessive generation of highly reactive free radicals—largely due to hyperglycemia—induces oxidative stress, which further exacerbates the development and progression of diabetes and its complications. Overproduction and/or insufficient clearance of these free radicals results in vascular dysfunction and further damage to cellular proteins, membrane lipids and nucleic acids. Several mechanisms are implicated in association with diabetes with oxidative stress. The generation of reactive oxygen species (ROS) follows glucose auto-oxidation in the presence of iron [3]. ROS-mediated stress is involved in various malfunctions and diseases, including immunodeficiency, pulmonary diseases and cardiovascular pathology [4]. This leads to protein glycation, overproduction of superoxide radicals at the mitochondria and increased nicotinamide adenine dinucleotide phosphate (NADPH) oxidase levels. NADPH is a membrane-bound enzyme complex that faces the extracellular space [5] and contributes to the elimination of invading pathogens in macrophages and neutrophils and thereby serves as an inflammatory mediator [6]. In DM, hyperglycemia is the main factor that induces oxidative stress, mainly via activation of NADPH oxidase in monocytes [7]. In type 2 DM, the susceptibility of β-cells to oxidative damage by NADPH oxidase has been related to β-cell dysfunction and provides clues on the therapeutic benefits of NADPH oxidase inhibitors for the prevention of metabolic stress associated with β-cell failure [8].

The field of antidiabetic medication in HF has changed radically since the development of new glucose-lowering agents and the numerous studies performed to determine their safety for widespread use [9]. The purpose of this article is to review the current literature on the use of antidiabetic agents and the role of effective glycemic control in patients with HF.

## 2. Materials and Methods

A literature search was conducted on the PubMed, MedLine and Embase databases and Google and updated on 15 January 2022 with the keywords “Metformin”, “Thiazolidinediones”,”Sulfonylureas”, “Meglitinides”, “Alpha glucosidase inhibitors”, “Insulin”, “Dipeptidyl peptidase 4 inhibitors”, “Sodium-glucose cotransporter-2 (SGLT-2) inhibitors”, “Glucagon-likepeptide-1 (GLP-1) receptor agonists”, “Oxidative stress”, “Heart failure with preserved ejection fraction (HFpEF)”, “Heart failure with reduced ejection fraction (HFrEF)”. Clinical trials regarding the action of these medications on HFpEF and HFrEF and their impact on the oxidative stress in diabetic patients were also analyzed. To be included in this review, articles were required to have been published before the end of January 2022 and have their full text available in English. They were further categorized as original research, reviews, meta-analyses or letters to the editor. Following database screening, titles and abstracts were reviewed to verify the inclusion criteria. An additional literature search was conducted for related references included in the manuscripts. The suitability of the scanned abstracts was assessed by two independent individuals so as to be included in this narrative review.

## 3. Epidemiology and Pathophysiological Pathway

Type 2 DM has evolved into a global epidemic and is expected to affect over 592 million people worldwide by 2035; in the United States, approximately 30.2 million adults (12.2% of the population) presented diabetes in 2015 [10]. Type 2 DM constitutes approximately 90–95% of all diabetic cases, and a majority of trials have focused on its pharmacological treatment [11]. Diabetes is independently associated with greater risk of death and rehospitalization compared to nondiabetic patients with HF. This fact is supported by many studies, such as the Swedish Heart Failure Registry [12], where mortality increased by 37% in cases of coexistence of diabetes and HF; in older adults enrolled in the Medicare program, mortality was 32.7 per 1000 person-years compared to 3.7 per 1000 person-years among those without HF. The Framingham Heart Study suggests that diabetes alone increases the risk of HF up to two-fold in men and five-fold in women compared to same-age matched controls [13]. 

Many epidemiological studies have shown a direct correlation between diabetes and HF, but pathophysiological changes that lead to such an increase in mortality rates are yet to be elucidated [11]. It seems that the coexistence of coronary artery disease, left ventricular hypertrophy and the diabetic cardiomyopathy as an entity can lead to the remodeling of cardiomyocytes and cardiac tissues. Additionally, the development of left ventricular dysfunction, as well as the increased ventricular pressures in an already stiffened ventricle, can an play important role in the pathophysiology of this entity. One of the characteristics of a diabetic heart is the concentric hypertrophy associated with reduced systolic strain. The main underlying mechanisms of ventricular hypertrophy are myocardial triglyceride deposition and/or increased extracellular volume, exacerbated by hyperinsulinemia due to insulin resistance, which are followed by collagen deposition and fibrotic tissue formation. This condition is also amplified by microvascular damage caused by the deposition of advanced glycation end products [1]. The advanced glycation end products operate like pattern recognition receptors of the immune globulin family, initiating inflammatory signaling, apoptosis, fibrotic remodeling and immune cell infiltration. Moreover, the increased levels of circulating glucose and fatty acids result in alterations in myocardial energy mechanisms and consequently lead to inappropriate lipid deposition on the cardiac tissues. As a result, cardiomyocytes store larger amounts of lipids, which eventually lead to myocardial cell damage due to lipotoxicity. The rest of the lipid fragments that are insufficiently processed activate the inflammatory signaling pathways—including protein kinase C and nuclear factor K, common signal pathways with insulin—resulting in fatty acid oxidation. In the meantime, the excessive caloric supply and the sufficient energetic demand lead to a functional impairment of oxidative phosphorylation [1]. The endothelial dysfunction caused by the metabolic syndrome is associated with myocardial dysfunction due to repeated episodes of vasoconstriction and reperfusion injury as well as increased small vessel permeability, interstitial edema, fibrosis and myocardial ischemia [11]. Finally, diabetic cardiomyopathy is associated with mechanical dysfunction, electrophysiologic abnormalities, subcellular disorders and catecholamine receptor downregulation due to abnormal elevated catecholamine changes in myocardial calcium transport and contractile proteins [11].

## 4. Oxidative Stress in Diabetic Cardiomyopathy and β-Cell Dysfunction

There is growing evidence that oxidative stress is involved in the pathophysiology and development of diabetic cardiomyopathy. It has been proposed that downregulation of peroxisome proliferator-activated receptor α (PPARα) induces dysregulation of nitrogen oxides (NOX) proteins, which are predominant isoforms expressed in cardiac tissue and contribute to the development of myocardial hypertrophy Figure 1 [14].

Experiments with diabetic rats have shown that NOX2 and NOX4 proteins are also associated with cardiac hypertrophy and fibrosis, while mice subjected to high fat diets demonstrated elevated NOX2 proteins associated with increased cardiomyocyte size [15,16]. Furthermore, in diabetic myocardium, the overexpressed myocardial kinase β isoform of protein kinase C (PKCβ) is accompanied by increased upregulation of pro-oxidant enzyme NADPH oxidase, which is a major upstream moderator of oxidative stress. Interestingly, it has been proposed that inhibition of PKCβ can attenuate myocardial hypertrophy [17]. Oxidative stress is closely related to diabetic complications that are responsible for both mortality and long-term disability in patients with diabetes. Sustained hyperglycemia in DM leads to increased ROS production by enhancing mitochondrial oxygen consumption and damaging mitochondrial function. Increased generation of ROS, reduced activity of endogenous antioxidants or both can induce oxidative stress, which is a potent culprit in diabetes mellitus due to inducing β-cell dysfunction and insulin resistance.

The impaired insulin secretion that is associated with overstimulation of β-cells by chronic hyperglycemia or free fatty acids (FFA) is the central mechanism of DM. The β-cell dysfunction is the result of the following [18]: Subnormal expression of antioxidants (SOD, CAT, GPx) in β-cells due to hyperglycemia, high FFA-induced ROS and reactive nitrogen species (RNS) accumulation [19].Chronic exposure of β-cells to oxidative stress inhibits insulin secretion by opening ATP-sensitive K+ channels and suppressing calcium influx, which results from the ROS-induced overproduction of cyclin-dependent kinase inhibitor p21 [20].Chronic exposure of β-cells to elevated FFA, which decreases mitochondrial membrane potential and leads to uncoupled protein-2 accumulation, can also activate β-cell ATP-sensitive K+ channels to inhibit insulin production [21].Reduced transcriptional activity of insulin genes by nuclear accumulation of pancreas duodenal homeobox factor 1, which is a key transcription factor responsible for maintaining β-cell function) by oxidative stress [22].Oxidative-stress-induced c-Jun N-terminal kinase activation and enhanced nuclear translocation of forkhead box protein O1, a key driver of metabolic disease that can suppress the binding of pancreatic and duodenal homeobox 1 to deoxyribonucleic acid (DNA) [23,24].Excessive long-chain acyl CoA will be generated in the process of increased β-cell fatty acid metabolism, which can keep β-cell ATP-sensitive K+ channels open to suppress ATP generation and insulin secretion [25].ROS overproduction reduces insulin secretion by suppressing the expression of MaFA, a member of the fundamental leucine zipper family of transcription factors involved in the transcription of insulin genes [22].Chronic-oxidative-stress-induced inflammation and nuclear transcription factor NF-kB activation through IL-1R signaling promote the expression of the Bcl-2 family of proapoptotic members [26,27], leading to β-cell damage and even apoptosis.

## 5. Oxidative-Stress-Mediated Cardiac Hypertrophy/Heart Failure and Diabetes

Oxidative-stress-induced-cardiac hypertrophy, fibrosis/apoptosis and further cardiac dysfunction/heart failure constitute a complex phenomenon involving several oxidative stress pathways, such as: a. ROS-generating enzymes (increased NOX protein expression), b. metabolic disorder in the context of diabetes (downregulation of CTRP9 genes, decreased AMPK phosphorylation, increased lipid kinases as PIKfyve, reduced mitochondrial antioxidant levels of thioredoxin 2), c. mitochondrial dysfunction (increased TRPC3 expression, increased mitochondrial iron transporting protein SFXN1, increased NCOA4 levels, reduced ENDOG levels, MTG1 deficiency, TIM50 deficiency), d. inflammation (increased toll-like receptor 4/TLR4 signaling, reduced Nrf2 transcription factor, FNDC5 deficiency, decreased hepatokine FGF21 levels) as well as e. dysregulated cellular processes (CDC20 up regulation, increased REGγ, and over expression of heat shock protein Hsp22) [15].
NOX proteins: ROS-generating enzymes

NOX proteins are known to produce ROS in a regulated manner. NOX2 and NOX4 isoforms are mainly expressed in cardiac tissue. Increased NOX2 protein expression has been correlated with cardiac hypertrophy and dysfunction development via downregulation of PPARa, action that could be reversed with fenofibrate (a PPARa activator) or NOX2 inhibitors in in vitro and in vivo mice models [14].

Similarly, elevated NOX4 levels in response to hypertrophic stimuli have been demonstrated to induce oxidative stress and cardiac dysfunction via either mitochondrial dysfunction due to ROS production or mitochondrial protein cysteine oxidation [28,29]. Additionally, increased NOX4 levels due to hypertrophic stimuli, such as angiotensin II, can further induce HDAC4 nuclear export, which plays a critical role in the suppression of cardiac hypertrophy [30], while DPP4 inhibitor teneligliptin was revealed to decrease cardiac hypertrophy via inhibition of NOX4 mRNA increase and HDAC4 export by increasing GLP1 levels [30]. In addition, NOX4 can promote volume overload via inactivation of protein phosphatase 2A(PP2A) and further increase Akt phosphorylation and modulation of RPS6 and 4E-BP1 downstream proteins, whereas NOX4−/− mice demonstrated less eccentric LV remodeling and attenuated cardiac hypertrophy due to reduction in phosphorylated Akt levels [31]. Conversely, NOX4 can preserve myocardial capillary density, offering protection against pressure overload [32] and promoting preferentially fatty-acid-oxidation, enhancing myocardial bioenergetics in cases of increased pressure overload [33]. Overexpression of calcium-regulated NOX5 isoform has been demonstrated to induce cardiac hypertrophy, fibrosis and dysfunction in transgenic NOX5 mice, worsening heart failure [34]. This was reversed with either ROS inhibitor (N-acetylcysteine) or L type calcium blocker diltiazem administration, rendering NOX5 the key mediator in ROS and calcium signaling pathways.

LOX, a copper-dependent amine oxidase that controls matrix remodeling, has been correlated with enhancement of cardiac hypertrophy and cardiac dysfunction in mice overexpressing human LOX as a consequence of excessive inflammatory response and ROS production via activation of p38 MAPK and simultaneous inhibition of AMP-activated protein kinase (AMPK) [35].
b.Metabolic disorders: obesity and diabetes

Both diabetes and obesity have been shown to play critical roles in the progression of cardiac hypertrophy and further cardiac dysfunction via multiple oxidative stress pathways’ activation. Mice fed with high fat diets revealed increased NOX2 levels correlated with elevated Akt and Erk1/2 phosphorylation, increased cardiomyocyte size and oxidative stress enhancement [36]. Additionally, NOX2−/− mice have demonstrated improvements in cardiac systolic function under pressure overload (REF new page 4), rendering NOX2 expression a common factor in both obesity- and pressure-induced cardiac hypertrophy. Oxidative stress induction via high-fat diet was shown to activate the BCL10/CARD9/p38 MAPK axis as well [37], demonstrating the signaling intermediate relation between oxidative stress and cardiac hypertrophy inducers. AMPK regulates lipid metabolism and maintains redox balance via induction of antioxidant genes (FOXO transcription factors) and attenuation of mTOR signaling [38]. In particular, downregulation of CTRP9 gene leads to reduced AMPK phosphorylation and increased mTOR phosphorylation, inducing cardiac hypertrophy, fibrosis, apoptosis and oxidative stress in the context of obesity induced cardiac hypertrophy [39]. On the contrary, treatment with CRTP9 protein enhances LKB1 phosphorylation and further AMPK activation, demonstrating antilipotoxic action via the LKB1/AMPK axis [40]. In another study, palmitate treatment can enhance mitochondrial fission and ROS generation via reduction of AMPK expression, inhibiting mitofilin interaction with both SAM50 and CHCHD3 (MICOS complex) [41], while acetylcholine can reduce cardiac hypertrophy via mitofilin expression and further AMPK activation. PIKfyve lipid kinase has been proven to increase mitochondrial ROS production and apoptosis, while its inhibition can attenuate oxidative stress and mitochondrial damage, resulting in reduced cardiac hypertrophy and improved cardiac function via a SIRT3-dependent pathway [42].

Hyperglycemia has been correlated with cardiac hypertrophy through oxidative stress induction, as H9c2 cell incubation in high-glucose has resulted in increased cell size and hypertrophy along with elevated ROS production [43] via reducing the levels of thioredoxin 2, a mitochondrial antioxidant that attenuates oxidative stress in both H9c2cells and diabetic rat myocardium [44]. A PPARd agonist, GW0742, has been shown to reduce hyperglycemia-induced cardiac hypertrophy via ROS production suppression and further attenuation of Erk1/2 and PI3K/Akt signaling in case of hypertrophic stimuli [45]. In another study, diabetic rats were reported to develop cardiac hypertrophy and further fibrosis analogous to high levels of NOX2 and NOX4 [46]. Additionally, advanced glycation end products (AGEs) are regarded as key factors between diabetes and oxidative stress, as some studies reveal oxidative stress to induce AGE formation, while others revealed AGEs to be mediators of oxidative stress [47]. In particular, in diabetic rats treated with metabolic modulator trimetazidine, diabetic cardiomyopathy was prevented by inhibition of NOX2/TRPC3 without affecting plasma AGE levels [48]. Similarly, isosteviol sodium therapy was demonstrated to enhance cardiac remodeling via inhibition of diabetes-induced Erk and NF-kB signaling pathways without any changes in AGEs, suggesting their upstream role in oxidative stress [49]. Additionally, metformin by normalizing glucose uptake and decreasing fatty acyl carnitines, prevents cardiac hypertrophy and improves cardiac function. Positive effects that have been attributed to AMPK and mTOR signaling pathway normalization include improved fatty acid oxidation and oxidative stress reduction [50]. Finally, calorie restriction has revealed a protective role against adverse cardiac remodeling in diabetic mice models via iron homeostasis normalization and further inhibition of oxidative stress and inflammation [51]. Additionally, calorie restriction reduces cardiac hypertrophy via mitoKATP activation and further intracellular redox balance improvement [52]. In clinical practice, caloric restriction has been revealed to reduce LV mass and thickness in older individuals with HF with preserved ejection fraction [53].
c.Mitochondrial dysfunction

Mitochondria play a critical role in cellular redox balance maintenance, as electron transport chain complexes constitute a major source of ROS generation during ATP production, while disturbances in mitochondrial antioxidant properties have been correlated with cardiac hypertrophy. Indeed, in patients with hypertrophic cardiomyopathy, upregulated levels of mitochondrial complex I contributed to increased hyperactivity and further increased ROS levels [54]. High-salt diet has been demonstrated to augment cardiac mitochondrial TRPC3 expression, leading to ROS production, mitochondrial dysfunction and cardiac hypertrophy [55], whereas TRPC3 deficiency inhibited cardiac hypertrophy and improved mitochondrial function. In addition, apelin-13, a ligand for the G-protein-coupled apelin receptor, can promote mitochondrial iron production by increasing levels of the iron-transporting protein SFXN1 and ROS production, leading to cardiac hypertrophy progression [56]. Mitochondrial DNA damage with age has been related to cardiac hypertrophy/dilatation and further systolic and diastolic dysfunction via oxidative stress induction [57]. Deficiency in ENDOG, which encodes a mitochondrial nuclease, has been correlated with ROS production via altering the Akt/GSK3b and class II HDAC signaling cascades and cardiac hypertrophy development [58]. Meanwhile, humanin treatment could restore ROS levels and reduce cardiac hypertrophy in ENDOG-deficient cardiomyocytes. MTG1, which regulates mitochondrial translation, has been shown to inhibit cardiac hypertrophy via reduction in ROS generation and downregulation of TAK1, p38 MAPK and Jnk1/2 stress signaling pathways in the context of pressure overload [59]. Additionally, TIM50 downregulation and deficiency have been associated with induction of cardiac hypertrophy and fibrosis via the ASK1/Jnk/p38 MAPK axis, while antioxidant NAC can reverse cardiac hypertrophy, suggesting a relation between TIM 50 and oxidative stress [60].
d.Inflammation

Inflammation has been correlated with myocardial fibrosis, cardiac hypertrophy and diastolic heart failure [61]. A significant link between inflammation, oxidative stress and cardiac hypertrophy constitutes toll-like receptor 4 (TLR4) signaling [62]. In particular, enhancement of TLR4 signaling has been reported to trigger ROS production, inflammation and CD68+ macrophage infiltration, leading to cardiac hypertrophy, fibrosis and HF, whereas administration of TLR4 antagonist prevented cardiac hypertrophy and dysfunction [63]. Additionally, increased TLR4 expression has been associated with mitochondrial dysfunction in an isoproterenol-induced cardiac hypertrophy rat model [64] as treatment with a TLR4-agonist, LPS, induced oxidative stress accelerating cardiac disease.

Nrf2 transcription factor deficiency has been reported to induce oxidative stress, inflammation, cardiac hypertrophy/fibrosis and further dysfunction via the IL-6/STAT3 axis [65]. In an attempt to maintain Nrf2 signaling, antioxidant peroxiredoxin 1 upregulation led to suppression of inflammation and oxidative stress via the Nrf2/HO-1 axis [66]. Notably, though levels of peroxiredoxin 1 were inherently upregulated in the context of pressure overload, they were proven insufficient to prevent cardiac hypertrophy, implying additional signaling pathways in this cascade.

Finally, adipokines, myokines and hepatokines can play a critical role as inflammatory mediators in cardiac hypertrophy. CTRP3 overexpression induced by ROS has been associated with aggravation of cardiac hypertrophy and function in murine hypertrophic hearts and failing human hearts. CTRP3 signaling is mediated via a TAK1/Jnk axis that was initiated by PKA [67]. FNDC5, by releasing the myokine irisin into the bloodstream, enhances cardiomyocyte metabolism, promoting mitochondrial biogenesis. FNDC5 deficiency promotes high-cholesterol-diet-induced cardiac hypertrophy, elevated inflammatory cytokine expression and oxidative stress via the JAK2/STAT3 pathway [68]. 

On the contrary, treatment with the FGF21 hepatokine is correlated with cardiac hypertrophy and fibrosis attenuation, apoptosis and cardiac dysfunction inhibition via increased deacetylase activity of SIRT1, promoting its further interaction with LKB1 and FOXO1 [69].
e.Dysregulated autophagy and protein homeostasis

Dysregulated autophagy, a vital cell process that preserves cell survival, is a well-recognized mechanism of cardiac hypertrophy. Interestingly, autophagy can be both upstream and downstream of oxidative-stress-mediated cardiac hypertrophy, as its dysregulation could result in defective cellular component accumulation. Oxidative stress can impair autophagy via p38 MAPK and Jnk signaling pathways, as their subsequent inhibition leads to autophagy restoration and further cardiac hypertrophy prevention [70]. CDC20 (anaphase-promoting complex activator) upregulation has been demonstrated in cases of cardiac hypertrophy. CD20 downregulation has been associated with reduction of NOX4 and ROS levels and further cardiac hypertrophy inhibition, whereas ectopic CDC20 expression enhanced cardiac hypertrophy via direct degradation of the autophagy regulator LC3 and further autophagy impairment [71].

Protein homeostasis maintenance is vital a cellular process for normal cell function and aberrant protein elimination. REGγ, a member of the 11S proteasome activator, can bind and further activate 20S proteasome, promoting the degradation of several proteins. REGγ upregulation has been correlated with FOXO3a nuclear export, with further MnSOD reduction and ROS increase in the context of pressure overload [72], while treatment with a MnSOD mimetic, MnTBAP inhibited ROS accumulation and cardiac hypertrophy. In another study, overexpression of the small heat shock protein Hsp22 in mice was correlated with elevated ROS production via enhanced activity of NADPH oxidase, xanthine oxidase and mitochondrial complex I, as well as further cardiac hypertrophy induction—action that could be reversed with the antioxidant tempol that, in turn, reduces cardiac hypertrophy [73].

## 6. Anti-Diabetic Drug Categories and Their Action on Oxidative Stress

### 6.1. Metformin

One of the well-established agents in diabetes treatment is metformin, which lowers blood glucose levels through a decrease in hepatic gluconeogenesis, intestinal reabsorption of insulin and the improvement of sensitivity to insulin. It remains the suggested first-line medication in patients with diabetes mellitus as it is a low-cost drug with a safe profile [9,13]. The role of metformin in HF was controversial during the years, as it was initially contraindicated in patients with HF due to the possibility of lactic acidosis induction [13]. Due to the lack of convincing evidence, in 2006, the FDA withdrew the contraindication of administration except for patients with advanced HF, chronic kidney disease (CKD) or both [13]. Paradoxically, the effects of metformin on the cardiovascular system are unproven. A recent post hoc analysis of the SAVOR-TIMI 53 trial (Assessment of Vascular Outcomes Recorded in Patients with Diabetes Mellitus-Thrombolysis in Myocardial Infarction) demonstrated the safety of saxagliptin; patients (n = 12,156) were classified as having taken versus never having taken metformin during the trial period [74]. Of the 12,156 patients, 8971 (74%) had received previously metformin, and 1611 (13%) had prior HF [74]. Metformin use was associated with no difference in the risk for the composite endpoint of cardiovascular death, myocardial infarction (MI) or ischemic stroke (HRIPTW 0.92; 95% CI 0.76–1.11) but with lower risk of all-cause mortality (HRIPTW 0.75; 95% CI 0.59–0.95) along with the interesting conclusion that there was no significant relationship between metformin use and these end points in patients with prior HF [74]. However, all cohort and observational studies have shown a reduction in the use of metformin as a first-line treatment in patients with DM and HF [9]. Another recent meta-analysis by Zhu et al. [75] showed that metformin had no significant difference in cardiovascular outcomes, but there was a possible decrease in the risk of major adverse cardiovascular events compared to placebo or no treatment (RR 0.84; 95% CI 0.71–1.00). Finally, the recommendation for metformin use in diabetic patients with HF is to be considered if the glomerular filtration rate (GFR) is stable and >30 mL/min/1.73 m^2^ (class IIa, level C) [76]. Khan et al. investigated the association between the introduction of metformin and sulfonylurea in the therapeutic management of patients with comorbid HF and DM [77]. The authors evaluated patients admitted for HF in the Get With The Guidelines-Heart Failure Registry between 2006 and 2014 presenting with a medical history of DM and no prescription of metformin or sulfonylurea before admission, and the patients were separately analyzed in groups with newly prescribed therapy with metformin and sulfonylurea and without it within 90 days of discharge. The administration of metformin was independently associated with reduced risk of composite mortality and hospitalization for HF (HR: 0.81; 95% CI: 0.67–0.98; *p =* 0.03) in 12-month clinical outcomes, driven by findings among patients with EF > 40%, whereas individual components were not statistically significant. The administration of sulfonylurea was associated with increased risk of mortality (HR: 1.24; 95% CI: 1.00–1.52; *p =* 0.045) and hospitalization for HF (HR: 1.22; 95% CI: 1.00–1.48; *p =* 0.050) with nominal statistical significance regardless of EF (all *p* for interaction > 0.11) [77]. Halabi et al. performed a systematic review and meta-analysis to investigate the association between the EF and mortality outcomes in HF/DM patients treated with metformin [78]. Four studies were meta-analyzed. Metformin was associated with an 18%-reduced mortality in HF with both preserved and reduced EF, especially in HF patients treated with therapies such as angiotensin-converting enzyme inhibitors (ACEi) and beta blockers (β = −0.2 [95% CI −0.3 to −0.1], *p* = 0.02). Greater protective effects were seen in patient group with EF > 50% (*p* = 0.003). A significant reduction in mortality has been observed in DM/HF patients treated with metformin along with insulin, ACEi and beta blocker therapy (insulin *p* = 0.002; ACEi *p* < 0.001; beta-blocker *p* = 0.017).

A recent experimental study [79] demonstrated that metformin can prevent methylglyoxal-induced apoptosis by suppressing oxidative stress in vitro and in vivo. Methylglyoxal is an active metabolite of glucose and plays a prominent role in the pathogenesis of diabetic vascular complications, including endothelial cell apoptosis induced by oxidative stress. Specifically, metformin prevents the apoptotic signaling cascades that are initiated by methylglyoxal-generated ROS by modulating the PI3K/Akt and Nrf2/HO-1 signaling pathways (intracellular signaling pathways important in regulating the cell cycle). Moreover, whereas metformin was not independently associated with clinical outcomes in patients with type 2 DM and heart failure with preserved ejection fraction (HFpEF), it was associated with lower all-cause mortality in the subgroup of patients with poor glycemic control [80]. In a recent report dealing with diabetic patients with advanced heart failure with reduced ejection fraction (HFrEF), patients treated with metformin demonstrated better quality of life and improved outcomes compared with the patients not receiving metformin [81]. The authors concluded that metformin should stay among frontline drugs for treatment of HFrEF patients with DM [81].

### 6.2. Thiazolidinediones

The next category of antidiabetic drugs currently approved by the FDA, thiazolidinediones pioglitazone and rosiglitazone—also known as “glitazones”—were initially promising antidiabetic factors [13]. They act as peroxisome proliferator-activated g receptor (PPAR)-g activators in adipose, muscle and liver tissue, resulting in a decrease in glucose production and subsequent increase in glucose utilization [13]. The first three randomized controlled trials after the approval of these drugs were DREAM (Diabetes Reduction Assessment with Ramipril and Rosiglitazone Medication) [82], PROactive (Prospective Pioglitazone Clinical Trial in Macrovascular Events) [83], and GSK211 (rosiglitazone) [13]. DREAM reported that the incidence of hospitalization for HF was 0.5% for patients treated with rosiglitazone in comparison to 0.1% in the placebo arm (HR 7.04; 95% CI, 1.60–31.0) [13,82]. The PROactive study included patients with known cardiovascular disease and revealed an increase of about 50% in hospitalization for HF in patients treated with pioglitazone (odds ratio [OR] 1.49; 95% CI, 1.23–1.80) [13,76,83]. The GSK211 trial randomized patients with NewYork Heart Association class I and II heart failure to rosiglitazone or placebo and showed significant increases in diuretic use and edema but failed to prove deterioration of left ventricular dysfunction [13]. The Thiazolidinediones Or Sulfonylureas and Cardiovascular Accidents Intervention Trial (TOSCA.IT), a large, randomized, unblinded trial studying pioglitazone vs. sulfonylureas add-on to treatment with metformin was stopped prematurely because of extraction of similar results for the composite endpoint between the two groups [76,84]. Τhe IRIS (insulin resistance intervention after stroke) trial demonstrated that pioglitazone reduced the combined endpoint of recurrent stroke and MI by 24% vs. placebo over a median follow-up of 4.8 years [13,76,85]. In the IRIS trial of 3876 insulin-resistant subjects without diabetes, the 5-year pioglitazone fracture risk was 13.6% compared with the placebo control of 8.8% with an HR of 1.53 [86]. All these randomized trials were followed by a meta-analysis by Zhu et al. [13,76], which showed increased risk of hospitalization for HF (RR 1.72; 95% CI, 1.21–2.42; *p* = 0.002), and as a result, since 2008, the use of thiazolidinediones has been contraindicated in diabetic patients with HF. Moreover, it has been shown that the risk for MI for rosiglitazone (RR 1.28; 95% CI 1.02–1.62) as well as for HF was increased (1.72, 1.31–2.27), whereas pioglitazone decreased the risk of major adverse cardiovascular events (RR 0.84; 95% CI 0.74–0.96), myocardial infarction (0.80; 0.67–0.95) and stroke (0.79; 0.65–0.95) but increased the risk of HF (1.40; 1.16–1.69) [75]. In conclusion, thiazolidinediones (pioglitazone and rosiglitazone) are not recommended for diabetic patients at high risk for HF as it is has been proven that they increase the risk of HF in this group of patients (class III, level A) [76].

Whereas pioglitazone, via the PPAR-γ/PGC-1α signaling pathway, reduces mitochondrial ROS production; inhibits oxidative stress and inflammation; improves mitochondrial biogenesis, dynamics, and function and induces atrial reverse remodeling that results in reduction in the incidence of inducible AF in diabetic rabbits [87], rosiglitazone increases inflammation and oxidative stress in patients with newly diagnosed T2DM [88].

### 6.3. Sulfonylureas

There are seven types of sulfonylureas (glimepiride, glipizide, tolazamide, tolbutamide, glyburide, chlorpropamide) that lower blood glucose levels by stimulating insulin release from the beta cells. They stimulate insulin release by blocking adenosine triphosphate (ATP)-sensitive potassium channels in the beta cells, reducing potassium permeability. This causes depolarization of the cell and increases calcium entry, increasing insulin secretion. The question of whether the sulfonylureas are safe for the cardiovascular system was first made controversial in 1971 after the publication of the University Group Diabetes Program (UGPD) [75]. During previous years, one large trial was dedicated to the effects of the use of sulfonylureas in cardiovascular patients: the CAROLINA (Cardiovascular Outcome Study of Linagliptin versus Glimepiride in Patients with Type 2 Diabetes) trial, which randomized patients with type 2 DM at high risk for atherosclerotic cardiovascular disease between a dipeptidyl peptidase (DPP)-4 inhibitor, linagliptin, and glimepiride (a second-generation sulfonylurea) and showed no difference for the primary end point of major adverse cardiovascular events (MACE) [13]. A previous meta-analysis based on observational studies revealed an increase in cardiovascular events with sulfonylureas’ usage, but given the inferiority of these studies due to their observational nature, newer meta-analyses based on updated cardiovascular trials reported that sulfonylureas may have no effect on cardiovascular events. However, glipizide and glimepiride can increase the risk of cardiovascular disease and the risk of stroke, respectively [75]. The most interesting conclusion of these meta-analyses is that cardiovascular mortality can differ between the sulfonylureas, with glicazide having lower cardiovascular mortality. As a result, the use of these agents in DM and HF should be personalized [76]. As far as oxidative stress is concerned, Glimepiride and Glipizide treated groups of gestational female Sprague Dawley rats weighing between 120–160 g showed statistically significant (*p* = 0.05) improvement in oxidative stress markers, blood glucose level, body weight, hematological parameters and lipid profile [89].

### 6.4. Meglitinides

This antidiabetic drug category, which enhances pancreatic insulin production, is not so well-studied [38]. Only one large trial, the NAVIGATOR (Nateglinide And Valsartan in Impaired Glucose Tolerance Outcomes Research) showed no reduction in major cardiovascular event with nateglinide, with no specific reference to HF patients [75,76,90]. Early studies, however, have shown that nateglinide lowers blood glucose, reduces insulin resistance and oxidative stress and improves endothelial function in newly diagnosed diabetes [91].

### 6.5. Alpha Glucosidase Inhibitors

Alpha glucosidase inhibitors (AGIs) act by altering the intestinal absorption of carbohydrates through inhibition of their conversion into simple sugars (monosaccharides) and thus decrease the bioavailability of carbohydrates in the body, significantly lowering blood glucose levels. The three AGIs used in clinical practice are acarbose, voglibose and miglitol. Alpha-glucosidase inhibitor acarbose, specifically, slows the digestion of starch in the small intestine, and as a result, the glucose enters the bloodstream more slowly [13]. The only large trial studying acarbose was the ACE (Acarbose Cardiovascular Evaluation) trial, which randomized 6522 patients to acarbose vs. placebo, with a primary endpoint of major adverse cardiovascular events (MACE) as well as hospitalization for unstable angina or HF [13,76]. This study showed no significant difference any-cause death, cardiovascular death, fatal or nonfatal MI, fatal or nonfatal stroke, hospital admission for unstable angina or hospitalization for HF [13,92]. Another study, STOP-NIDDM (Study to Prevent Non-Insulin-Dependent Diabetes Mellitus) randomized 1429 patients with impaired glucose tolerance between receiving acarbose or placebo and suggested a decreased risk of cardiovascular outcomes in general, but the number of patients with reported cardiovascular events was low. Therefore, no safe conclusions can be drawn [75,93]. There is experimental evidence that acarbose reduces fat deposition in liver tissue and decreases the size of adipose tissue cells and total antioxidant capacity [94].

### 6.6. Insulin 

Observational studies have shown an increase in cardiovascular mortality and presence of HF in patients treated with insulin, which is in part expected as these patients are usually older with greater risk for HF [10]. Some trials have studied the effect of insulin in HF: the ORIGIN trial (Outcome Reduction with Initial Glargine Intervention) [95] and the DEVOTE trial (A Trial Comparing Cardiovascular Safety of Insulin Degludec Versus Insulin Glargine in Subjects with Type 2 Diabetes at High Risk of Cardiovascular Events) [96].

The ORIGIN trial [95] examined insulin glargine versus standard care in patients with type 2 DM and high risk for cardiovascular disease and revealed that the addition of basal insulin has a neutral effect on the cardiovascular outcomes in comparison with the standard care group. Nonsignificant difference was observed by the use of insulin in HF hospitalization [95]. 

The second main study, the DEVOTE trial, randomized diabetic patients in high cardiovascular risk to be treated with degludecor with insulin glargine in order to examine the efficacy and safety of the former [96]. No significant difference was proven between the two insulins considering the development of adverse cardiovascular events, even in presence of heart failure [96].

The conclusion of the above trials is that the use of insulin does not have a significant effect on cardiovascular outcomes in high-risk diabetic patients [10]. Therefore, in the 2019 guidelines for the management of DM, insulin may be considered for patients with advanced systolic HFrEF with class IIb, level C evidence [76].

In a recent report [97], insulin therapy in HF patients with DM was associated with a higher mortality risk than oral hypoglycemic agents alone, regardless of the patients’ left ventricular ejection fraction and HF etiology. In patients with low HbA1c levels or more severe forms of heart failure, insulin therapy has been found to be harmful. The authors of this report suggested that specific management strategies and blood glucose targets may be needed when using insulin in patients with HF. A recent study [98] revealed that plasma nitric oxide, which is an indicator of oxidative stress, was significantly higher in a subgroup of young adults with type1DM and daily insulin treatment when compared to healthy age-adjusted controls. Since DM is mainly associated with insulin resistance and decreased insulin sensitivity due to impaired insulin signal transduction, current efforts are focused on potential substances to ameliorate oxidative stress and insulin resistance [99]. 

### 6.7. Dipeptidyl Peptidase 4 Inhibitors

Dipeptidyl peptidase-4 (DPP4) is a protein in humans that is encoded by the DPP4gene. It is an integral membrane protein expressed on cells throughout the body and can cleave a large number of bioactive molecules, including incretin hormones. DPP4 inhibitors prolong and enhance the activity of incretins, which are hormones which play an important role in insulin secretion from β-cells and blood glucose control regulation [10]. There are five DPP4 inhibitors in use in daily clinical practice: alogliptin, linagliptin, saxagliptin, sitagliptin and vidagliptin, and respectively, there are five large prospective trials in patients with DM at different risks for MACE that evaluated the association of these drugs’ administration with the development of cardiovascular events: SAVOR TIMI-53 (Saxagliptin Assessment of Vascular Outcomes Recorded in Patientswith Diabetes Mellitus thrombolysis in myocardial infarction) [74,100], EXAMINE (Examination of Cardiovascular Outcomes with Alogliptin versus Standard of Care) [101], TECOS (Trial Evaluating Cardiovascular Outcomes with Sitagliptin) [102], CARMELINA(Cardiovascular and Renal MicrovascularOutcome Study With Linagliptin in Patients With Type 2 Diabetes) [103] and CAROLINA [10,76,104]. In recent years, many preclinical studies have suggested probable cardiovascular benefit resulting from the use of these drugs. However, the above-mentioned trials identified that treatment with DPP4 inhibitors does not improve cardiovascular morbidity and mortality in comparison with placebo [56]. In particular, SAVOR-TIMI 53 demonstrated the negative effect of saxagliptin on hospitalization for HF in comparison with the placebo group [10,76,100]. As a result of the above trial, saxagliptin is not recommended in diabetic patients with HF (class III, level B) [76]. EXAMINE showed the neutral effect of alogliptin in HF hospitalization [10,76,101], and so did the trials TECOS [94] for sitagliptin and CARMELINA for linagliptin [10,76,103]. Only the CAROLINA study revealed similar cardiovascular safety for linagliptin and glimiperide [105]. The data from the above trials led the investigators to suggest, in the recent guidelines, that sitagliptin and linagliptin may be considered class IIb, level B in patients with type 2 DM and HF because of their neutral effect on the risk of developing HF [76]. Despite that, DPP4 inhibitors’ exact mechanisms in heart failure are not well studied; a recent study from Sano et al. [105] reported that a possible mechanism could be the sympathetic activation by DPP4 which increases the risk of HF. This could explain the neutral effect of sitagliptin and alogliptin, which are mainly excreted in the urine and suppress renal sodium-hydrogen exchanger 3 activity [105].

A recent observational study that assessed several biomarkers of oxidative damage in propensity-score-matched cohorts, including ROS, plasma advanced glycation end products, advanced oxidation protein products, carbonyl residues and the ferric-reducing ability of plasma and leukocyte DNA oxidative damage in Fpg sites, does not suggest any major effect of DPP4 on oxidative stress in humans [106].

### 6.8. Sodium–Glucose Cotransporter-2 (SGLT-2) Inhibitors

It is undeniable that two new categories of oral antidiabetic factors from the last decade (Table 1), sodium-glucose cotransporter-2 (SGLT-2) inhibitors—or gliflozins—and glucagon-likepeptide-1 (GLP-1) receptor agonists, seem to reduce cardiovascular risk significantly [107]. As a result of the development of these drugs, a plethora of trials focused on the cardiovascular outcomes of the administration of these factors [69]. Two main sodium-glucose cotransporters exist in the human body: SGLT1 and SGLT2. SGLT2 are found in renal tissue in the brush border of the renal tubular cells in the first segments of the proximal tubules, whereas SGLT1, aside from the kidney, are mostly found in the small intestine, heart and skeletal muscle. In the kidneys, SGLT2 and SGLT1 handle sodium and glucose reabsorption in the proximal tubules of the nephron. Their function is to reabsorb 100% of filtered glucose, leaving no glycosuria. Their capacity to transfer glucose is high but the affinity is low and are responsible for the reabsorption of 90 to 97% of filtered glucose. The remaining 3–10% of filtered glucose is absorbed by the high-affinity, low-capacity transporter SGLT1. The SGLT-2 inhibitors block renal glucose reuptake in the renal tubules, and as a result, they promote loss of glucose in the urine and improve blood pressure through urinary glucose and sodium excretion [76,108]. However, the basic mechanisms of action do not explain the impact on the cardiovascular system, and therefore, it is suggested that there are more complex mechanisms involved in this process [108].

Natriuresis induced by SGLT-2 inhibitors leads to an increase in the activation of the renin–angiotensin–aldosterone system (RAAS) and a reduction in plasma volume, systemic blood pressure and vascular stiffness via a decrease in the sympathetic system hyperactivation, resulting in increased vasodilatation and improved vascular function. This has also been demonstrated by dapagliflozin’s effectiveness on vascular endothelial function and glycemic control (DEFENCE study) via the use of dapagliflozin on early-stage DM type 2 patients, possibly via antifibrotic mechanisms [108,109,110,111]. Natriuresis and osmotic diuresis also increase glomerular afferent arteriolar vasoconstriction, and as a result, intraglomerular hyperfiltration is reduced, leading to decreased proteinuria and increased glycosuria [109]. In glycosuria, the uric acid in plasma is reduced, as are blood glucose levels and potential hyperglycemia-associated toxicity, especially in hyperglycemic patients [110]. The production of advanced glucose end (AGE) products and receptors for AGE (RAGE) products and the levels of HbA1c are reduced, while the production of glucagon is increased; along with the decrease in the insulin levels, this leads to fewer hypoglycemic episodes and therefore to better glycemic control [109,110]. Body weight and visceral and epicardial fat are reduced [109], and the production and metabolism of ketone bodies are increased, resulting in an improvement in myocardial energy efficiency and a reduction of liver steatosis [108,109,110]. SGLT-2 inhibitors are associated with the development of euglycemic ketoacidosis, primarily with the triggering of circulating ketones as a result of glucose level decrease; furthermore by stimulating lipolysis, which is accompanied by an increase in free fatty acids (FFA). That contributes to ketogenesis along with reduced urinal excretion of ketones [112]. The beneficial effects of SGLT2-i due to ketogenesis were described in a pig model of HF, where the administration of empagliflozin improved left ventricular remodeling and systolic function by enhancing cardiac energy [113]. Obesity and T2DM can lead to HFpEF due to an increase in cardiac preload as a result of plasma volume expansion. Hypertrophic visceral adipocytes release proinflammatory cytokines, which lead to arterial stiffness and endothelial dysfunction in the arterioles and a reduction in capillary density, leading to an increase in cardiac afterload [112]. Along with a reduction in erythropoiesis and the hematocrit levels, it is estimated that the use of SGLT-2 inhibitors in patients with DM type 2 with HF reduces the risk of HF decompensation and hospitalization via decrease in preload, afterload, left ventricle wall stress, myocardial hypertrophy, cardiac muscle demands on O_2_ as well as via cardiac fibrosis inhibition and the prevention of the development of arrhythmiogenic mechanisms [109,110,114]. Natriuresis and osmotic diuresis regulated by SGLT-2i result in an improvement of both preload and afterload, preserving the heart rate and reducing arterial stiffness. Hyperglycemia is also associated with the myocardial damage caused by inflammation, increased ROS, fibrosis and apoptosis due to the formation of nonenzymatic glycation end products of proteins, lipids and nucleic acids [115]. A mice model has demonstrated that SGLT-2 inhibition reduces the circulating levels of chemokine 2, IL-6 and TNF-a as well as NF-kB in renal tissue and n CRP in hepatic cells and adipocytes [116,117]. Adipokines in obesity and T2DM are altered, introducing a proinflammatory state. Leptin favors cardiovascular diseases, and adiponectin is associated with a more cardioprotective role. Alterations in leptin levels cause the accumulation of epicardial fat and subsequently ventricular remodeling, cardiac fibrosis and inflammation [118]. SGLT-2i reduces serum leptin and increases adiponectin concentrations favoring cardioprotection, affecting IL-6, TNF-a and PAI-1 variably [112]. The overexpression of nitric oxide synthase (iNOS) leads to a reduction in the activities of two proteins: an isoform of the binding protein X-box 1 (XBP1) and the enzyme 1α. A reduction in XBP1 expression inhibits the protein response and therefore results in the accumulation of destabilized proteins and products of cardiomyocyte apoptosis [119]. Another target of SGLT2i is AMP-activated protein kinase (AMPK), a regulator of metabolic homeostasis which promotes catabolism and inhibits anabolism as a mediator of several signaling hormones with protective effects over mitochondria, inflammation, apoptosis and fibrosis [120]. TGF-β/Smad, which is strongly involved in tissue fibrosis regulations, is another signaling pathway which can be altered by empagliflozin, which induces its blockage and therefore a decrease in the fibrotic transformation of myocardial tissue [121]. SGLT-2i can lead to a significant loss of glucose in urine, and subsequently, the glycosuria activates the enzymes sirtuin 1 (SIRT1) and adenosine monophosphate-activated protein kinase (AMPK), which have antioxidant and anti-inflammatory effects and upregulate proliferator-activated receptor-γ coactivator-1α (PGC-1 α) [108,110,121].The SIRT1/AMPK/PGC-1 α signaling path induces autophagic clearance of destroyed mitochondria and promotes the biogenesis of healthy ones, thereby muting inflammation [121]. A 2021 study investigated the mechanisms through which empagliflozin affected HFpEF in human and murine hearts. The study came to the conclusion that the administration of empagliflozin reduces inflammatory and oxidative stress in HFpEF and thereby improves the NO–sGC–cGMP–cascade and protein kinase G Ia (PKGIa) activity through reduced PKG-Ia oxidation and polymerization, eliminating pathological cardiac stiffness [122,123].

### 6.9. Studies on the Effect of SGLT2 Inhibitors on Cardiovascular Outcomes and Heart Failure 

According to recent multiple studies (Table 2), SGLT-2 inhibitors can lower the risk of HF hospitalization, and they have been recognized to exert multidimensional effects since approval of their usage in the United States [114,124]. The three SGLT-2 inhibitors (ipragliflozin, tofogliflozin and luseogliflozin) have only been approved in Japan [124]. There are four major cardiovascular outcome trials (CVOTs) about SGLT2 inhibitors: Empagliflozin Cardiovascular Outcome Event Trial in Type 2 Diabetes Mellitus Patients Removing Excess Glucose (EMPA-REG OUTCOME) [125,126,127,128], the EMPRISE (empagliflozin comparative effectiveness and safety) [129], the impact of empagliflozin on cardiac function and biomarkers of heart failure in patients with acute myocardial infarction (EMMY) trial [130], the Empagliflozin Outcome Trial in Patients with Chronic Heart Failure with Preserved and Reduced Ejection Fraction (EMPEROR-PRESERVED [131] and EMPEROR-REDUCED [132]), EMPIRE-HF (Empagliflozin in heart failure patients with reduced ejection fraction) [133], the Canagliflozin Cardiovascular Assessment Study (CANVAS) Program [75,124,134,135], Comparative Effectiveness of Cardiovascular Outcomes in New Users of SGLT-2 Inhibitors (CVD-REAL) [121], the Canagliflozin and Renal Events in Diabetes with Established Nephropathy Clinical Evaluation (CREDENCE) trial [75,125] and Dapagliflozin Effect on Cardiovascular Events-Thrombolysis In Myocardial Infarction (DECLARE-TIMI 58) [75,136,137,138].

EMPA-REG OUTCOME randomized 7020 patients with chronic DM (57% > 10 years) and cardiovascular disease into two groups to be treated with either empagliflozin 10 or 25 mg o.d., or placebo [76]. It was obvious that empagliflozin led to a significant reduction in the three-point composite primary outcome (CV death, non-fatal MI or non-fatal stroke) by 14% compared to placebo and a main reduction of about 38% in CV death (*p* < 0.0001) within two months in the patients’ follow-up [39,75,76]. A secondary analysis of EMPA-REG OUTCOME strengthened the suggestion that empagliflozin was associated with a 35% reduction in hospitalization for HF (*p* < 0.002), again quite early in the follow-up period after the therapy initiation, resulting in a positive effect on HF decompensation risk [77,115,126]. Overall mortality was also reduced by 32% (*p* < 0.0001), a fact that seemed to be associated with a reduction in mortality associated with pump failure, while the arm of CV mortality related to atherosclerotic events presented no significant difference between the two subgroups [76,107].

A metanalysis on EMPA-REG OUTCOME by Savarese et al. showed a reduction in HF readmissions as well as in the composite outcome of HF readmissions and CV or all-cause mortality in patients treated with empagliflozin in the post-acute HF period [128]. Another metanalysis by Fitchett et al. [128] studied the positive effects of empagliflozin across the spectrum of baseline CV risk and revealed that regardless of the cardiovascular risk, the observed reduction in CV outcomes and mortality by empagliflozin was evident in all the subgroups of patients.

The EMPRISE (Empagliflozin Comparative effectiveness and Safety) study evaluated the effectiveness, safety and utilization of empagliflozin in routine healthcare [129]. The first interim analysis of this study dealt with the evaluation of the risk of hospitalization for HF complications in patients with DM receiving empagliflozin vs. those receiving sitagliptin. It showed that empagliflozin was superior to sitagliptin as an additional factor to standard therapy, and its use has also been associated with a decrease in the risk of hospitalization for HF decompensation regardless of history of CV disease [129].

The EMMY trial is a randomized, double-blind, placebo-controlled, phase 3b trial that included patients with acute MI and severe myocardial necrosis, who were randomized for either empagliflozin (10 mg once daily) or placebo, with primary endpoint changes in NT-proBNP within 6 months after acute MI. Secondary outcomes include hospitalization rates due to HF or other causes, duration of hospital stay and all-cause mortality [130]. EMMY will be the first trial testing empagliflozin in patients with MI regardless of their glycemic status, and its result is anticipated to be very helpful in the daily practice of the medical community [130].

The most valuable knowledge about empagliflozin comes from the EMPEROR-PRESERVED [131] and EMPEROR-REDUCED [132] trials [27,50]. The EMPEROR-PRESERVED trial included 5988 patients with class II–IV HF and an ejection fraction >40%, who received empagliflozin (10 mg once daily) or placebo beyond optimal medical therapy. The primary outcome was a composite of cardiovascular mortality or hospitalization hospitalization for HF adverse events [131]. The primary outcome event occurred in 13.8% of the patients in the empagliflozin group and in 17.1% in the placebo group (hazard ratio, 0.79; 95% confidence interval [CI], 0.69 to 0.90; *p* < 0.001), indicating a lower risk of hospitalization for HF in the empagliflozin group along with a higher percentage of uncomplicated genital and urinary tract infections and hypotension in the empagliflozin group. These beneficial events were consistent across patients with preserved ejection fraction regardless of the presence of DM.

The EMPEROR-REDUCED trial was the first randomized placebo-controlled trial that studied the impact of SGLT-2 inhibitors in the treatment of HF with reduced ejection fraction [132]. Patients with class II–IV HF and a left ventricular ejection fraction ≤40% were randomized to receive empagliflozin (10 mg daily) or placebo as an additional treatment factor to the standard medical therapy. The primary outcomes were cardiovascular death or hospitalization for HF complications, total hospitalizations for HF and adverse renal outcomes. The primary outcome rates were higher in patients with diabetes but similar between patients with prediabetes and normoglycemia. Empagliflozin reduced the risk of the primary outcome in all patients regardless of their glycemic status (hazard ratio, 0.72 (95% CI, 0.60–0.87) and 0.78 (95% CI, 0.64–0.97), respectively, *p*-interaction = 0.57), decreased the decline rate in renal function, prevented the development of serious renal events and led to significant improvement in the health status of diabetic and nondiabetic patients, without proof of reduced HbA1c or hypoglycemic episodes [132].

Another trial with promising future results is the EMPIRE-HF (Empagliflozin in heart failure patients with reduced ejection fraction) randomized, double-blinded, placebo-controlled, parallel group, clinical trial that randomized patients with HF with reduced left ventricular ejection fraction (≤40%) on optimal therapy, in either empagliflozin 10 mg daily or placebo with primary endpoint the effect on NT-proBNP [85]. The completion of the study will help the scientific community to understand the action of SGLT2 inhibitors in cardiac, renal, and/or metabolic mechanisms, and the impact of SGLT2 inhibitors on physical activity and quality of life [134].

The Canagliflozin Cardiovascular Assessment Study (CANVAS) Program integrated data from two randomized control trials (RCTs): CANVAS and CANVAS-Renal (CANVAS-R). This was a regulatory safety program of canagliflozin, the first FDA-approved SGLT-2 inhibitor, in patients with type 2 DM and cardiovascular disease or at high risk of cardiovascular disease [76,134]. The program randomized 10,142 patients with DM at high cardiovascular risk between canagliflozin 100 mg or canagliflozin 300 mg per os and placebo [75]. A reduction in composite three-point MACE of 14% (*p* = 0.02) was observed for canagliflozin without significant reduction in cardiovascular death or overall death, while positive results were observed for hospitalization for adverse HF events [76,124].

The CANVAS and CANVAS-R programs were followed by Figtree et al.’s [135] analysis, which evaluated the effects of canagliflozin use in HF patients with reduced vs. preserved ejection fraction that were enrolled in the above-mentioned trials. This analysis showed that canagliflozin reduced the risk of HF events in general in patients with type 2 DM regardless of the ejection fraction [135]. However, there is the limitation that the ejection fraction used for the patient categorization was evaluated at the time of the event and that it was not the baseline ejection fraction [135].

Later, the CREDENCE trial in June 2019 randomized 4401 patients with type 2 DM and albuminuric chronic kidney disease (eGFR 30 to <90 mL/min/1.73 m^2^) between canagliflozin and placebo, demonstrating a relative reduction in the primary renal outcome of 30% by canagliflozin after a median follow-up of 2.6 years [76,125]. A significant reduction of prespecified secondary CV outcomes of three-point MACE (HR 0.80, 95% CI 0.67–0.95; *p* = 0.01) and hospitalization for adverse HF events (HR 0.61, 95% CI 0.47–0.80; *p* < 0.001) was observed via the use of canagliflozin in comparison with placebo in this very-high-CV-risk group of patients [76,125]. All these trials are characterized by hospitalizations, and it could be assumed that the beneficial effects achieved in these trials are more likely the result of a reduction in HF-associated CV events [76].

In 2017, the Comparative Effectiveness of Cardiovascular Outcomes in New Users of SGLT-2 Inhibitors (CVD-REAL) trial retrospectively compared the effects of SGLT-2 inhibitors in cardiovascular morbidity and mortality against other antidiabetic factors, and it showed a reduction of 39% in hospitalizations for HF-adverse events in patients treated with SGLT-2 inhibitors [124]. There was proportional reduction in hospitalizations for HF and all-cause mortality (46%) in these patients [124]. One year later, the CVD-REAL 2 trial also reported reductions in hospitalization for HF adverse events and all-cause mortality of 36% and 40%, respectively, in patients treated with SGLT-2 inhibitors [124].

DECLARE-TIMI 58 evaluated the effect of 10 mg dapagliflozin per os vs. placebo in 17,160 patients with DM and cardiovascular disease or with multiple cardiovascular risk factors, and it did not prove inferiority of the composite three-point MACE [136]. Two following primary efficacy analyses showed that dapagliflozin did not significantly reduce MACE but resulted in a lower rate of the combined endpoint of cardiovascular mortality and hospitalization for HF complications (4.9 vs. 5.8%; HR 0.83, 95% CI 0.73–0.95; *p* = 0.005) [124,136]. The benefit of dapagliflozin was also proportional in the subgroup with cardiovascular disease or with multiple CV risk factors [76].

A subanalysis of the DECLARE-TIMI 58 trial demonstrated that cardiovascular mortality, as well as HF-associated hospitalizations, were more frequent in patients with type 2 DM and prior MI, and dapagliflozin reduced the relative risk of MACE by 16% and the absolute risk by 2.6 % in these patients (15.2% versus 17.8%; hazard ratio [HR], 0.84; 95% CI, 0.72–0.99; *p* = 0.039), whereas there was no effect in patients without previous MI (7.1% versus 7.1%; HR, 1.00; 95% CI, 0.88–1.13; *p* = 0.97) [136]. The relative risk reductions in cardiovascular death/hospitalization for heart failure were similar, but the absolute risk reductions tended to be greater: 1.9% vs. 0.6% in patients with and without previous MI, respectively [136].

Verma et al. [115] evaluated the effects of dapagliflozin on type 2 DM patients with or without MI and HF based on two large analyses of the DECLARE-TIMI 58 trial; dapagliflozin demonstrated a greater reduction in HF complications or cardiovascular mortality in patients with HF with reduced ejection fraction (≤45%) in comparison to patients without reduced ejection fraction (HR 0.62 vs. 0.88; *p*-interaction = 0.046) [115]. The interesting fact is that hospitalizations due to adverse HF events were affected positively by the use of dapagliflozin in all patient groups [115].

Another metanalysis from Kato et al. analyzed the results of DECLARE-TIMI 58, evaluating the impact of baseline ejection fraction on the clinical benefits using SGLT-2 inhibitors [137]. The investigators came to the conclusion that the use of dapagliflozin leads to a reduction of hospitalizations for HF regardless of the presence of reduced ejection fraction, but the reduction of cardiovascular mortality and all-cause mortality was observed only in patients with reduced ejection fraction [137].

Another meta-analysis by Zenliker et al. [138] reassessed the known beneficial impact of SGLT-2 inhibitors on HF hospitalizations and cardiovascular mortality, as well as on renal disease progression, irrespective of previous atherosclerotic cardiovascular disease or a history of heart failure; the reduction in MACE was only apparent in patients with established cardiovascular disease.

Another systematic meta-analysis of randomized controlled and observational studies [139] demonstrated a significant decrease in hospitalizations for HF in patients treated with SGLT-2 inhibitors compared to placebo or other antidiabetes drugs for type II diabetes. (OR 0.70, 0.64, 0.66, respectively, *p* = 0.000).

The dapagliflozin in patients with HFrEF (DAPA-HF) trial evaluated patients with chronic symptomatic HF with reduced ejection fraction, randomizing them at a ratio of 1:1 between 10 mg dapagliflozin once daily and placebo, and the follow-up observation was of a total duration of 18 months [140]. Heart failure deterioration occurred in 16.3% of patients treated with dapagliflozin vs. 21.2% in the placebo group, while death from cardiovascular causes occurred in 9.6% in the dapagliflozin group vs. 11.5% in the placebo group. Therefore, the positive effect of dapagliflozin was obvious, as the primary composite endpoint of cardiovascular death or hospitalization or urgent visit for adverse HF events was reduced by 26% [140,141]. Interestingly, dapagliflozin contributed to significant improvement in the recurrence of HF-adverse events, health status, development of HF associated symptoms and all-cause mortality [140,141]. Dapagliflozin was beneficial regardless of the presence of diabetes, which was observed in about 55% of patients enrolled in the study, with a reduction in serious adverse events (including volume depletion, acute kidney injury and hypoglycemia) being described in all patient groups [140,141]. DAPA-HF revealed the improvement of patients’ clinical outcomes and health status by dapagliflozin in the treatment of HF with reduced ejection fraction [92]. Approximately 11% of patients in DAPA-HF were already treated with an angiotensin receptor neprilysin inhibitor (ARNI), and the addition of dapagliflozin in the previous therapeutic profile was associated with beneficial effects. This has particular importance as there are no interactions between the two drug categories [140]. The results of DAPA-HF and the beneficial effects of dapagliflozin in nondiabetic patients with HF were proven and reinforced by the study on SGLT-2 inhibitors, as was their therapeutic role in other cardiometabolic conditions, apart from their already-known positive glycemic effects in type 2 DM [140]. The positive effects of dapagliflozin regardless of the presence of DM may have to do with the improvement of glycemic status as well as with the impact of ketogenesis in HF patients [10].

A subsequent metanalysis of DAPA-HF revealed the positive effect of dapagliflozin use in diabetic patients with HF with reduced ejection fraction on the composite outcome of reduced cardiovascular death, improved health status and quality of life [142]. Martinez et al. [143] analyzed the impact of dapagliflozin in this group of patients based on their age, and this meta-analysis consolidated the positive effect of dapagliflozin in the reduction of cardiovascular mortality and adverse HF events regardless of age, confirming the safety of dapagliflozin use even in elderly patients.

The DEFINE-HF (Dapagliflozin Effects on Biomarkers, Symptoms and Functional Status in Patients with HF with Reduced Ejection Fraction) trial randomized 263 patients with HF with reduced ejection fraction (<40%) between dapagliflozin 10 mg daily and placebo for 12 weeks. The primary outcomes were a ≥5-point increase in HF disease-specific health status on the Kansas City Cardiomyopathy Questionnaire overall summary score, or a ≥20% decrease in NT-proBNP [144]. There was no association between the use of dapagliflozin and the reduction in NT-proBNP levels, but a greater proportion of patients showed clinically improvement by the use of dapagliflozin versus placebo (61.5% vs. 50.4%, respectively) regardless of the presence of DM [144].

Additionally, in 2019, a Scandinavian-register-based cohort study compared the use of SGLT-2 inhibitors to the use of DPP4 inhibitors and showed a reduction in HF-associated and any-cause mortality as well as in cardiovascular events in patients treated with SGLT-2 inhibitors [145].

The Evaluation of Ertugliflozin Efficacy and Safety Cardiovascular Outcomes trial (VERTIS-CV) revealed no superiority for 3-point MACE for cardiovascular mortality, nonfatal MI or stroke with ertugliflozin, but the reduction in hospitalization for heart failure adverse effects rate was significant compared to placebo (2.5 vs. 3.6%, respectively) [108]. This fact was associated with a possible difference between the studied populations of the conducted trials; for example, 10% of patients in the EMPA-REG trial had underlying HF conditions, while the corresponding rate in VERTIS-CV was 23.1% [108].

A common characteristic of the above studies was the fact that the reduction in cardiovascular mortality and HF hospitalization was observed early in the treatment period, with SGLT2 inhibitors suggesting their potential beneficial cardiovascular effects in patients without DM [126].

As a novel class of glucose-lowering agents, SGLT-2 inhibitors can reduce high-glucose-induced oxidative stress and act as indirect antioxidants. These agents, by scavenging free radicals and boosting biological antioxidant systems, seem to guide novel therapeutic strategies not only for diabetes and cardiovascular diseases but also for nephropathies, liver diseases, neural disorders and cancers [146]. Indeed, SGLT-2 inhibition may limit glucose uptake, resulting in energetic crisis, post oxidative stress mediated DNA damage and cell cycle arrest, resulting in increased cell apoptosis and decreased proliferation of thyroid cancer cells [147]. By promoting autophagy, mitochondrial biogenesis and attenuation of oxidative stress, inflammation and apoptosis, empagliflozin has already improved renal ischemia/reperfusion injury in nondiabetic rats [148]. Empagliflozin alone or combined with other antidiabetics has ameliorated hepatic steatosis, inflammation, oxidative stress and fibrosis in mouse models with non-alcoholic fatty liver disease [149].

### 6.10. Glucagon-Likepeptide-1 (GLP-1) Receptor Agonists

The other category of antidiabetic drugs—Glucagon-like peptide-1 (GLP-1) receptor agonists, also called incretin mimetics—are agonists of the GLP-1 receptor and increase insulin secretion and inhibit glucagon release [95]. The hypoglycemia mechanism in patients treated with GLP-1 agonists is shown in Figure 2.

The most important studies about these agents are ELIXA (Evaluation of Lixisenatide in Acute Coronary Syndrome) [150], LEADER (Liraglutide and Cardiovascular Outcomes in Type 2 Diabetes) [125,151], SUSTAIN-6 (Trial to Evaluate Cardiovascular and Other Long-term Outcomes with Semaglutidein Subjects with Type 2 Diabetes) [152], EXSCEL (Exenatide Study of Cardiovascular Event Lowering) [153], Harmony Outcomes (Albiglutideand CV outcomes in patients with type 2 diabetes mellitus and cardiovascular disease) [154], REWIND (Researching Cardiovascular Events With a Weekly Incretin in Diabetes) [155] and PIONEER 6 (Peptide Innovation for Early Diabetes Treatment) [156].

In particular, the ELIXA trial demonstrated that lixisenatide’s action was equal to placebo in a four-point MACE (three-point MACE and additionally hospitalization for unstable angina) in patients with diabetes post-MI [76,150]. There were no significant differences between the two groups in hospitalization rates for HF-adverse events (hazard ratio in the lixisenatide group, 0.96; 95% CI, 0.75 to 1.23) or the morality rates (hazard ratio, 0.94; 95% CI, 0.78 to 1.13) [150].

The LEADER trial studied 9340 patients with diabetes and high risk of cardiovascular events treated with liraglutide vs. placebo as an extra drug in their standard antidiabetic therapy. This study supported the beneficial impact as there was a significant reduction in the composite primary endpoint (CV mortality, nonfatal MI, or nonfatal stroke) by 13% as well as reductions in cardiovascular mortality and total mortality by 22 and 15%, respectively [76,125,151]. However, there was no significant reduction observed in the rates of nonfatal MI, nonfatal stroke and HF-related hospitalization [85].

In addition, in the SUSTAIN-6 study [104], semaglutide reduced the three-point MACE and nonfatal stroke by 26% and 39%, respectively [76,152]. No superiority was shown between semaglutide and placebo regarding HF-associated hospitalization [152].

In the EXSCEL study, a 10% relative risk reduction for MACE (HR 0.90, 95% CI, 0.8160.999; nominal *p* = 0.047) was proven in the subgroup of patients with cardiovascular disease [39,76]. Hospitalization for HF was reduced in the exenatide group versus placebo (HR 0.82, 95% CI 0.68–0.99; *p* = 0.038) [153]. In a recent prespecified analysis of EXSCEL, it was proven that exenatide had a positive effect in all-cause mortality and first hospitalization for HF adverse events only in patients with baseline HF [153].

In the Harmony Outcomes trial, albiglutide led to a significant reduction of 22% in three-point MACE as well as a reduction in MI of 25% compared to placebo, but the rates of hospital admissions for HF did not significantly differ between the two groups [154].

The REWIND trial [155] studied the effect of a once-weekly subcutaneous dose of dulaglutide (1.5 mg) vs. placebo on three-point MACE in patients with DM with a previous history of cardiovascular event or at high risk for a cardiovascular event. A total rate of 12.0% of patients treated with dulaglutide had a primary composite outcome event vs. 13.4% in the placebo group (HR 0.88, 95% CI 0.79099; *p* = 0.026), and as a result, a cardiovascular superiority in 3-point MACE was proven, but the incidence of all-cause mortality, heart failure, revascularization and hospital admission was not affected [87,125,155].

The PIONEER-6 trial compared the effect of oral semaglutide o.d. (target dose 14 mg) vs. placebo on cardiovascular outcomes in patients with DM and high cardiovascular risk. The study demonstrated noninferiority for protection of cardiovascular events by the use of oral semaglutide [156]. It significantly reduced the risk for CV death at 0.9% vs. 1.9% in placebo and all-cause mortality at 1.4% in the semaglutide group vs. 2.8% in the placebo group. First hospitalization for HF events occurred in 1.3% of the patients in the semaglutide group and 1.5% in the placebo group [156].

A recent meta-analysis of these studies by Kristensen et al. showed that GLP-1 receptor agonists reduced MACE by 12% (HR 0.88, 95% CI 0.82–0.94; *p* < 0.0001), with a HRs 0.88 (95% CI 0.81–0.96; *p* = 0.003) for mortality due to cardiovascular causes, 0.84 (0.76–0.93; *p* < 0.0001) for fatal or nonfatal stroke and 0.91 (0.84–1.00; *p* = 0.043) for fatal or nonfatal MI. There was also a reduction in all-cause mortality by 12% (0.88, 0.83–0.95; *p* = 0.001) and hospital admission for HF by 9% (0.91, 0.83–0.99; *p* = 0.028 [104].

Based on the above evidence, GLP1-receptor agonists (lixisenatide, liraglutide, semaglutide, exenatide and dulaglutide) may be considered for patients with DM and HF as class IIb, level of evidence A [76]. The results of the above trials were used by the American Diabetes Association and the European Association for the Study of Diabetes in a consensus report, and it was suggested that in the treatment of patients with DM and cardiovascular disease, a glucagon-like peptide-1 (GLP-1) receptor agonist or a SGLT2 inhibitor should be used as an extra treatment in standard therapy with metformin [157]. In the group of patients with chronic kidney disease or atherosclerotic cardiovascular disease and HF, an SGLT-2 inhibitor should be preferred [157]. Moreover, these agents improve serum levels of antioxidant biomarkers, and this improvement is even more prominent when used in combination with similar drugs, suggesting a synergistic and additive action [158].

## 7. Conclusions

The evidence that supports the beneficial use of antidiabetic drugs in the treatment of diabetic patients with HF as an additional therapeutic mechanism is gradually growing. The current class I recommendation for the treatment of HFrEF is the use of inhibitors of the neprilysin or renin–angiotensin systems, β-blockers and mineralocorticoid receptor antagonists [132]. An innovative addition in the treatment of HF would be the additional use of dapagliflozin considering the results from DAPA-HF trial [99]. The majority of antidiabetic medications apply favorable action and reduce oxidative stress. Despite that these drugs, in mono and/or combined therapy, are able to provide good glycemic control, they have failed to show consistent reduction in cardiovascular mortality. Therefore, future efforts should elucidate potential mechanisms that play a major role, such as hyperglycemia- and hyperlipidemia-evoked oxidative stress, mitochondrial dysfunction and inflammation. Further data on the beneficial mechanisms of SGLT-2 inhibitors in the treatment of HF regardless of the presence of DM or the baseline ejection fraction are expected to be extracted by future clinical trials [132].

## 8. Strengths and Limitations

In this review, we attempt to summarize the current literature, interpolating well-established pathophysiological pathways (diabetic cardiomyopathy, nephropathy, heart failure) and molecular mechanisms (oxidative stress) into clinical trials and real-world data regarding all spectra of antidiabetic agents. However, the lack of a strict systematic search protocol or conduction of meta-analysis for each individual antidiabetic category constitute the main limitations of our study.

## Figures and Tables

**Figure 1 jcm-11-04660-f001:**
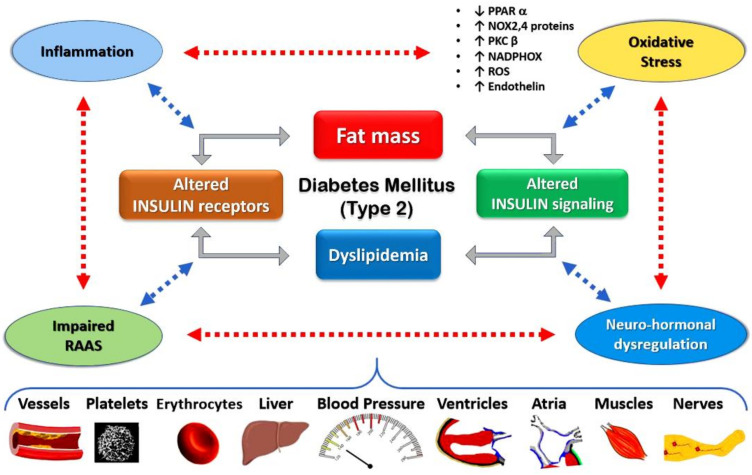
Pathophysiology of multiorgan damage in Type-2 Diabetes. Oxidative stress is involved in the pathophysiology and development of multiorgan disease inpatients with type 2 diabetes mellitus. It has been proposed that downregulation of PPARα (small down arrow) induces dysregulation (overproduction, small up arrow) of the NOX proteins, which are predominantly expressed isoforms in cardiac tissue and contribute to the development of myocardial hypertrophy and cardiac dysfunction. Increased vasoconstriction and peripheral arteriolar resistances also result from the activation of RAAS and neurohormonal dysregulation, often based upon overproduction of endothelin(s) (small up arrow) and reduced nitroxide synthase activity. Hyperglycemia activates most vascular stressors, which are responsible for micro- and macrovascular dysfunction and atherosclerotic complications in several districts. The overexpression of myocardial kinases β-isoform of PKC (small up arrow) is accompanied by upregulation of NAPHOX (small up arrow), one of the oxidative distressing mediators in patients with diabetic cardiomyopathy. Chronically higher glycemic levels also simulate ROS production (small up arrow) and increase mitochondrial O_2_ consumption. Oxidative stress also affects O_2_ conveyance by erythrocytes and promotes activation of platelets. Transport of O_2_ and nutrients to organs and tissues are damaged over time. For further explanation, see text. NADPHOX = NADPH oxidase; NOX2, 4 = nitrogen pxides 2, 4; PPAR α = peroxisome-proliferator-activated receptor α; PKCβ = protein kinase C isoform β; RAAS = renal–angiotensin–aldosterone system; ROS = reactive oxygen species.

**Figure 2 jcm-11-04660-f002:**
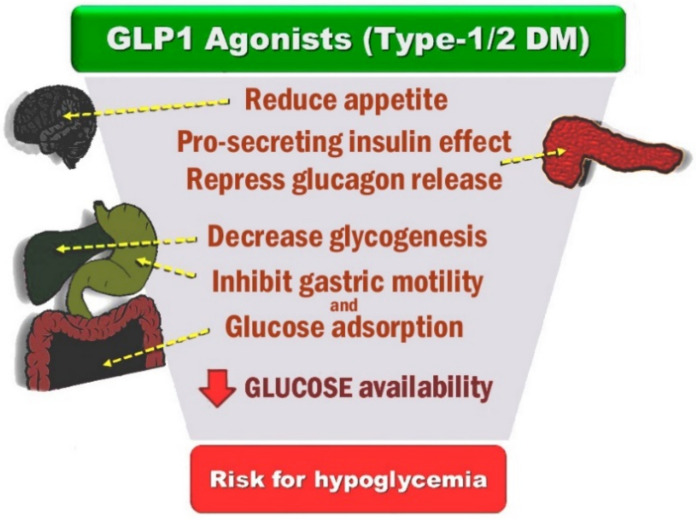
Glucagon-like peptide-1 (GLP-1) receptor agonists, also called incretin mimetics, are agonists of the Glucagon-like peptide 1 receptor that increase insulin secretion, inhibit glucagon release and improve serum levels of antioxidants.

**Table 1 jcm-11-04660-t001:** The sodium–glucose cotransporter-2 (SGLT-2) inhibitors and glucagon-likepeptide-1 (GLP-1) receptor agonists in everyday practice.

**Sodium–glucose cotransporter-2 (SGLT-2) inhibitors: Block renal glucose reuptake and promote loss of glucose in the urine, thus improving blood pressure via glucose and sodium excretion.**
Canagliflozin (Invokana) taken by mouth once daily Dapagliflozin (Farxiga, Forxiga) taken by mouth once daily Empagliflozin (Jardiance) taken by mouth once daily Ertugliflozin (Steglatro) taken by mouth once daily Ipragliflozin (Suglat) taken by mouth once daily Luseogliflozin (Lusefi) taken by mouth once daily Remogliflozin etabonate (Remo, Remozen) taken by mouth once daily Sotagliflozin (Zynquista) taken by mouth once daily Tofogliflozin (Apleway, Deberza) taken by mouth once daily
**Glucagon-like peptide-1 (GLP-1) receptor agonists: Increase insulin secretion and inhibit glucagon release via stimulation of the GLP-1 receptors**
Dulaglutide (Trulicity) taken by injection weekly Exenatide extended release (Bydureon) taken by injection weekly Exenatide (Byetta) taken by injection twice daily Liraglutide (Victoza) taken by injection daily Lixisenatide (Adlyxin) taken by injection daily Semaglutide (Ozempic) taken by injection weekly Semaglutide (Rybelsus) taken by mouth once daily

**Table 2 jcm-11-04660-t002:** Clinical trials studying the effect of SGLT2 inhibitors on cardiovascular outcomes and heart-failure-related events.

Trial	Medication Used	Results	Meta-Analyses
EMPA-REG OUTCOME [40,76,77,78,79,80,81]	Empagliflozin vs. placebo	- 14% reduction in composite CV death, non fatal MI, non fatal stroke - 38% reduction in CV death - 35% reduction in HF associated hospitalisation	Savarese et al.: reduction in HF-associated readmissions and all composite post-acute HF period Fitchett et al.: regardless of the CV risk, the positive effects concerned all groups of patients
EMPRISE [82]	Empagliflozin vs. sitagliptin	- empagliflosin was superior to sitagliptin in decreasing the hospitalization risk for HF decompensation regardless of CV disease history	
EMMY [83]	Empagliflozin vs. placebo	- about to test empagliflozin in patients with MI regardless of glycaemic status	
EMPEROR-PRESERVED [84]	Empagliflozin vs. placebo in patients with preserved ejection fraction	- lower risk of hospitalization for HF in the empagliflozin group - uncomplicated genital and urinary tract infections and hypotension in the empagliflozin group - benefit consistent across patients with preserved ejection fraction regardless of the presence of DM	
EMPEROR-REDUCED [85]	Empagliflozin vs. placebo in patients with reduced ejection fraction	- reduced risk of cardiovascular death or hospitalization for HF complications, total hospitalizations for HF, and adverse renal outcomes in all patients regardless of their glycaemic status	
EMPIRE-HF [86]	Empagliflozin vs. placebo in patients with reduced ejection fraction	- targets the effect on NT-proBNP and the impact on cardiac, renal and metabolic mechanisms; physical activity and quality of life	
CANVAS Programm [39,76,87,88]	Canagliflozin vs. placebo	- reduction at three point MACE by 14% in the canagliflozin group with positive effects in the hospitalisation rates and without significant results in mortality	Figtree et al.: canagliflozin reduced the risk of HF events in general in patients with type 2 DM, regardless of the presence of reduced or preserved ejection fraction
CREDENCE [39,77]	Canagliflozin vs. placebo	- relative reduction in the primary renal outcome of 30% by canagliflozin - significant reduction in prespecified secondary CV outcomes of three-point MACE and hospitalization for adverse HF events by the use of canagliflozin in comparison with the use of placebo in the very-high-CV-risk group of patients	
CVD REAL [76]	SGLT-2 inhibitors vs. other anti-diabetic factors	- 39% reduction in hospitalizations for adverse HF events - 46% reduction in all-cause mortality	
DECLARE-TIMI 58 [39,76,89,90,91]	Dapagliflozin vs. placebo	- lower rate of combined endpoint of cardiovascular mortality and hospitalization for HF complications	Furtado et al.: reduced risk of MACE in DM patients compared to MI patients, similar risk reduction in cardiovascular mortality/heart failure hospitalization with a greater absolute risk reduction estimated at 1.9% for patients with prior MI vs. 0.6% in patients without prior MI Verma et al.: dapagliflozin had a positive effect in hospitalizations due to HF averse events and led to a greater reduction in HF complications or cardiovascular mortality in patients with HF with reduced ejection fraction (≤45%) in comparison with patients without reduced ejection fraction Kato et al.: dapagliflozin leads to a reduction of hospitalizations for HF regardless of the presence of reduced ejection fraction; the reduction of cardiovascular mortality and all-cause mortality was observed only in patients with reduced ejection fraction
DAPA-HF [94,95,96,97,98]	Dapagliflozin vs. placebo	- positive effect of dapagloflozin in HF adverse events, health status, all cause mortality, regardless of the presence of DM	Kosiborod et al.: dapagliflozin improved health status and quality of life and reduced cardiovascular mortality Martinez et al.: safe use of dapagliflozin in elderly
DEFINE-HF [99]	Dapagliflozin vs. placebo	- no association between the use of dapagliflozin and the reduction in the NT-proBNP levels - clinical improvement by the use of dapagliflozin versus placebo regardless of the presence of DM	
VERTIS-CV [71]	Ertugliflozin vs. placebo	- no superiority for three-point MACE for cardiovascular mortality, non-fatal MI or stroke - significant reduction in hospitalization rates for HF adverse events	

## Data Availability

The study did not report new data.

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
