# Peer review of "Anti-Diabetic Therapy, Heart Failure and Oxidative Stress: An Update"

_jcm, 2022, doi:10.3390/jcm11164660_

Round 1

Reviewer 1 Report

1. Suggest including a more inclusive conclusion to highlight the overall directions of the current anti-diabetic therapies targeting heart failure. The current abstract focus on the recent highlights of the individual therapies.

2. Suggest having comprehensive description of the figure caption to encompass the overview on the inter-relationships between inflammation, oxidative stress, impaired RAAS and neuro-hormonal dysregulation on DM2

3. The following categories have included the overview on the mode of action for each drug classes and their recent updates from clinical trial findings. However, their action on oxidative stress remain under reported or only limited information were included. Suggest incorporating more comprehensive description on how these drug categories exert their action on oxidative stress. 

4. There is lack of info on the mechanism on oxidative stress in section 6.5 and 6.8; Section 6.9 have a disproportionate amount of information as compared to other sections. Suggest reducing or summarizing the overall info included

Reviewer 2 Report

The paper by Koniari is a review on the pharmacological treatment of patients with diabetes and heart failure.

For a better clarity I would suggest to include a paragraph describing the main oxidative pathways involved in diabetes and HF. Furthermore, the formation of RCS including methylglyoxal and glyoxal and protein cabonylation and how these reactions are involved in the pathogenesis and disease progression should be better detailed. 

A review on the most recent research reporting oxidative stress in patients with diabetes and HF should be added.

Graphs summarizing the action of each drug would make the paper more readable.

Minor

Typos Line 159

Legend (why here?) 162-164

Reviewer 3 Report

This is a very comprehensive review of the subject and the authors pulled together data from a wide range of trials and papers. I enjoyed reading this review, I have a few comments:

There is a slight over reliance on ref 1 and 11 in the major paragraph on page 3. I would like to see a more rounded discussion on the link between diabetes and the heart. 

Figure 1, erythrocytes rather than red cells. 

Section 4 page 3; Can you provide evidence for the relationship between the reduced ppar and dysfunction. We have classically seen an increase in PPARa target genes and proteins in diabetic hearts (rodent that is) and I would be interested to see the evidence for a downregulation.

Page 7: might be worth adding in a few more papers in the link between metformin and HFpEF, Halabi et Al 2020 would suggest that there is slightly more protection than the paper you linked here. Having said that there was a really interesting paper in JACC: Heart Failure in March 2022 (khan et al 2022) which looked at HFpEF and HFrEF

Section 6.8: there have been some nice mouse studies that have looked at the effects of sglt2 in the heart concluding the role of ketones and alterations in cardiac metabolism. It would be good to reflect on these mechanistic studies in this section, including several papers from Gary Lopaschuk’s group.
